



# NeuralMie (v1.0): An Aerosol Optics Emulator

Andrew Geiss[1] and Po-Lun Ma[1]

[1]Pacific Northwest National Laboratory, Richland, WA, USA

**Correspondence:** Andrew Geiss (andrew.geiss@pnnl.gov)

**Abstract.** The direct interactions of atmospheric aerosols with radiation significantly impact the Earth's climate and weather and are important to represent accurately in simulations of the atmosphere. This work introduces two new contributions to enable more accurate representation of aerosol optics in atmosphere models: 1) "TAMie," a new Python-based Mie scattering code that can represent both homogeneous and coated particles and achieves comparable speed and accuracy to established

Fortran Mie codes. 2) "NeuralMie," a neural network Mie code emulator trained on data from TAMie, that can directly compute the bulk optical properties of a diverse range of aerosol populations and is appropriate for use in atmosphere simulations where aerosol optical properties are parameterized. NeuralMie is highly flexible and can be used for a large range of particle types and wavelengths. It can represent core-shell scattering, and by directly estimating bulk optical properties, is more efficient than existing Mie code and Mie code emulators while incurring negligible error (0.08% mean absolute percentage error).

**1   Introduction**

Aerosols have a substantial impact on atmospheric radiation. They influence the Earth's radiative budget both directly (Hansen et al., 2005; Johnson et al., 2018) and through their impacts on clouds (Twomey, 1977; Albrecht, 1989; Fan et al., 2016). Consequently, accurately simulating aerosols and their interactions with the Earth system is critically important for weather and climate modeling. This is challenging to do, and aerosol direct and indirect effects are among the largest sources of internal

uncertainty in current climate projections (Bellouin et al., 2020; Boucher et al., 2013). While some difficulties modeling aerosol radiative effects stem from lack of knowledge (e.g. the limits of observational constraints and our understanding of aerosol-cloud interactions), even the components of the problem that might be considered "solved" from a knowledge standpoint often cannot be adequately simulated due to computational constraints. Physics at the scale of aerosol particles simply cannot be resolved in a global-scale atmosphere simulation, and representation of aerosols in these simulations, along with a host of

other physical processes, is usually enabled by a suite of parameterizations and simplified physical models (Neale et al., 2012) that trade off physical realism for computational tractability.

Recently, machine learning (ML), and neural networks in particular, have emerged as powerful tools for developing new, more accurate, faster, and more capable, physics parameterizations for atmosphere modeling (Rasp et al., 2018; Brenowitz and Bretherton, 2018; Boukabara et al., 2021; Krasnopolsky et al., 2013). Conventional parameterizations typically take the form

of simplified hand-derived physical models, basic statistical models like lookup tables and linear regressions, and sometimes simply rely on expert heuristics to make decisions about the behaviour of a system. Machine learning provides an exciting data-




driven alternative to this approach, and recent advances in our ability to train and deploy deep neural networks (Goodfellow et al., 2016) have significantly accelerated research in this area. Applications where a physical process is well understood, but computationally expensive to simulate directly, are well suited to machine learning emulators. In these cases, training data can
be generated using a model that accurately represents the physics underlying a system and a significantly faster ML model can be trained to emulate it.

This work focuses on developing a ML-based emulator for accurately estimating the optical properties of aerosol populations. It is intended for use alongside modal aerosol models (Whitby and McMurry, 1997; Liu et al., 2012, 2016; Wang et al., 2020). Modal aerosol models are designed to simulate aerosol populations with limited computational expense. They are ca-
pable of representing a variety of different aerosol species, and they simplify the modeling problem by assigning each species into a limited number of size modes. Each mode assumes that its constituent aerosol species have prescribed log-normal size distributions. When a climate or weather model simulates radiative transfer, the radiation code must be fed the optical properties of each model grid cell, and to do this, the aerosol optical properties (AOPs) of the simulated aerosol populations must be estimated. This is typically done with a parameterization (Ghan et al., 2001; Ghan and Zaveri, 2007) that estimates solutions
using Lorenz-Mie theory (Bohren and Huffman, 1983), and NeuralMie has been designed to fill this role, though it can also be used as a general purpose model for estimating the optical properties of log-normally distributed spherical particle populations. NeuralMie includes the capability to represent "core-shell" (also called "coated sphere") scattering (Toon and Ackerman, 1981). The core-shell model represents particles as concentric spheres composed of different aerosol species. Aerosol mixing assumptions can significantly alter the optical properties of particles. For black carbon, which is the strongest absorbing
aerosol species (Bond et al., 2013), representing mixed particles as an insoluble black carbon core coated in a shell of another material (e.g. sulfate) is more physically realistic and typically results in lower absorption than internal volume mixing, but higher absorption than external mixing due to the shell acting as a lens and focusing light onto the black carbon core (Bond and Bergstrom, 2006).

Several past studies have applied machine learning to aerosol scattering and optics. Chen et al. (2022) used a neural network
to represent scattering by spheroidal dust particles, Yu et al. (2022) trained a large neural network on a database of non-spherical particles to predict particle optics, and Ren et al. (2020) trained a neural network to predict information about aerosol size distributions from photometer observations of AOPs. Both Thong and Yoon (2022) and Stremme (2019) trained neural networks to directly emulate a Mie scattering model. Veerman et al. (2021) developed a machine learning emulator to compute gas optics, typically a component of an atmospheric radiative transfer parameterization. In a similar vein, there have been numerous
efforts to develop machine learning emulators for the radiation code typically used in climate models (Krasnopolsky et al., 2012; Pal et al., 2019; Liu et al., 2020; Belochitski and Krasnopolsky, 2021; Song and Roh, 2021; Ukkonen, 2022; Stegmann et al., 2022; Lagerquist et al., 2023), which is very computationally expensive despite relying on significant simplifications to the radiative transfer problem (Pincus and Stevens, 2013; Iacono et al., 2008; Mlawer et al., 1997; Mlawer and Clough, 1997). Though rather than directly ingesting information from the aerosol model, these ML radiative transfer emulators typically
ingest parameterized aerosol optical properties, assume climatological aerosol properties, or ignore aerosols altogether. Most closely related to this study is our past work developing an aerosol optics emulator specifically designed to replace the optics





parameterization in the Energy Exascale Earth System Model (E3SM) (Geiss et al., 2023). Here, we take a significant step beyond that work by introducing a model that is highly flexible, appropriate for use in other models and applications, and is capable of representing core-shell scattering.

This work provides two major contributions:

1. **"TAMie"** - a new Python-based Mie scattering code, based on Toon and Ackerman's 1981 algorithm, for modeling the optics of individual particles. It can simulate scattering by both homogeneous and coated spheres and provides greater stability and accuracy than existing options, while achieving comparable speed to Fortran Mie codes using numba compiling (Lam et al., 2015).

2. **"NeuralMie"** - a neural network based emulator for estimating the bulk optical properties of log-normally distributed aerosol populations. It supports a wide range of input particles, represents homogeneous or coated spheres, and works for any plausible wavelength and geometric mean radius combination by performing calculations with respect to the size parameter, making it wavelength (and thus radiation code and aerosol model) agnostic. Because it directly performs bulk calculations for entire particle size distributions, NeuralMie is extremely fast compared to Mie code and other
75        Mie optics emulators, and achieves negligibly small mean absolute percentage errors in mass extinction coefficients (0.05% and 0.08% for homogeneous and coated sphere cases respectively). It is suitable for use alongside any model that assumes log-normally distributed spherical aerosol populations.

The manuscript is broken into three main sections. We first discuss the optics of individual particles and describe TAMie (Section 2). Then we discuss how bulk AOPs are parameterized and introduce a re-formulation of the problem that allows for
training highly accurate and flexible neural networks (Section 3). Finally, we describe and evaluate NeuralMie (Section 4).

## 2   Particle Optics

Calculating the optical properties of aerosol populations first requires computation of the scattering properties of individual particles. Theoretical representations of light scattering by small spheres were found by Gustav Mie in 1908 (Mie, 1908) and independently by several other researchers (Horvath, 2009). The more complicated case of concentric spheres with different
refractive indices was solved later by Aden and Kerker (1951). For an individual homogeneous spherical particle the extinction ($Q_e$), scattering ($Q_s$), and absorption ($Q_a$) coefficients are:

$$Q_e = \frac{2}{x^2} \sum_{n=1}^{\infty} (2n+1)\mathrm{Re}\{a_n + b_n\} \tag{1}$$

$$Q_s = \frac{2}{x^2} \sum_{n=1}^{\infty} (2n+1)(|a_n|^2 + |b_n|^2) \tag{2}$$


$$Q_a = Q_e - Q_s \tag{3}$$





while the asymmetry parameter, which describes the mean of the cosine of the scattering angle and ranges from -1 to 1, is given by:

$$g = \frac{4}{x^2 Q_s} \sum_{n=1}^{\infty} \left[ \frac{n(n+2)}{n+1} \text{Re} \left\{ a_n a_{n+1}^* + b_n b_{n+1}^* \right\} + \frac{2n+1}{n(n+1)} \text{Re} \left\{ a_n b_n^* \right\} \right] \tag{4}$$

(Wiscombe, 1980). Here, $x$ is the size parameter, defined: $x = 2\pi r / \lambda$ where $r$ is the particle radius and $\lambda$ is the wavelength of the radiation interacting with the particle. $a_n$ and $b_n$ are known as the "Mie coefficients." They depend on $x$ and $m$ (the particle's complex refractive index) and can be expressed in terms of Ricatti-Bessel (RB) functions and their derivatives. The definitions of the Mie coefficients in terms of RB functions can be found in Bohren and Huffman (1983) for both homogeneous spheres (their Eqs. 4.56 and 4.57) and coated spheres (their Eq. 8.2).

While the theoretical Mie solutions have been known for some time, numerical difficulties arise when computing the Mie coefficients, and often a large number of Mie coefficients (100s) are needed for the summations in Eqs. 1-4 to converge, exacerbating the problem. Numerical difficulties can occur because the RB functions that define $a_n$ and $b_n$ and their logarithmic derivatives, are computed using recursion relations, and numerical solutions can explode due to small errors introduced by limited machine precision growing exponentially when a large number of terms are calculated. Wiscombe (1979) provides a

detailed discussion of how the necessary recursion relations can be computed safely for the case of homogeneous spheres. Toon and Ackerman (1981) introduced the first stable Mie code capable of representing arbitrary coated spheres. They solve the numerical stability problem by expressing $a_n$ and $b_n$ entirely in terms of ratios and products of RB functions and their logarithmic derivatives that do not explode when a large number of terms are computed, and then deriving recurrence relations so that those products and ratios can be calculated directly. In the next section, we describe a Python based implementation of

their algorithm.

### 2.1 The "TAMie" Python package

As a component of this study, we have written a new Python-based Mie code. The code is stable for a large range of input parameters, easy to read and use (compared to Fortran implementations), and by leveraging just-in-time compiling achieves comparable speed to Fortran Mie code. At the time of writing, Python Mie codes did exist, but they did not meet our exact

requirements in terms of speed and reliability. The PyMieScatt package (Sumlin et al., 2018) is popular within the atmospheric sciences, and we used it to calculate optics for homogeneous spheres in our previous work (Geiss et al., 2023), but PyMieScatt's coated sphere implementation uses interpreted Python and was too slow to generate the large volume of training data needed to train NeuralMie. Instead, we have produced our own Python core-shell Mie solver that implements the Toon and Ackerman (1981) algorithm. Of course, more sophisticated algorithms have since been developed, for calculating the optical properties

of particles composed of any number of layers for instance (Wu and Wang, 1991; Johnson, 1996), but that level of complexity is not required for our use case. We have slightly altered Toon and Ackerman's original work to express everything in terms of the dimensionless size parameter and added a recursion relation that they did not use (Eq. 14 (Shiloah, 2018)), so in this section we provide a full overview of the algorithm implementation to serve as a companion to the published code. The next





section (2.2) provides a thorough evaluation of the new Mie code against PyMieScatt (Sumlin et al., 2018), a Fortran algorithm
for coated spheres (Wiscombe, 1993), and a Fortran algorithm for homogeneous spheres (Bohren and Huffman, 1983).

The Mie coefficients for coated spheres are given by (Toon and Ackerman, 1981; Bohren and Huffman, 1983):

$$a_n = \left[\frac{\psi_n(x_s)}{\zeta_n(x_s)}\right] \frac{\left(\left[\frac{\psi'_n(m_s x_s)}{\psi_n(m_s x_s)}\right] - m_s \left[\frac{\psi'_n(x_s)}{\psi_n(x_s)}\right]\right)(m_c + UW) - U\left[\frac{\psi_n(m_s x_c)}{\psi_n(m_s x_s)}\right]^2}{\left(\left[\frac{\psi'_n(m_s x_s)}{\psi_n(m_s x_s)}\right] - m_s \left[\frac{\zeta'_n(x_s)}{\zeta_n(x_s)}\right]\right)(m_c + UW) - U\left[\frac{\psi_n(m_s x_c)}{\psi_n(m_s x_s)}\right]^2} \tag{5}$$

$$b_n = \left[\frac{\psi_n(x_s)}{\zeta_n(x_s)}\right] \frac{\left(m_s \left[\frac{\psi'_n(m_s x_s)}{\psi_n(m_s x_s)}\right] - \left[\frac{\psi'_n(x_s)}{\psi_n(x_s)}\right]\right)(m_s + VW) - m_s V\left[\frac{\psi_n(m_s x_c)}{\psi_n(m_s x_s)}\right]^2}{\left(m_s \left[\frac{\psi'_n(m_s x_s)}{\psi_n(m_s x_s)}\right] - \left[\frac{\zeta'_n(x_s)}{\zeta_n(x_s)}\right]\right)(m_s + VW) - m_s V\left[\frac{\psi_n(m_s x_c)}{\psi_n(m_s x_s)}\right]^2} \tag{6}$$

$$U = m_c \left[\frac{\psi'_n(m_s x_c)}{\psi_n(m_s x_c)}\right] - m_s \left[\frac{\psi'_n(m_c x_c)}{\psi_n(m_c x_c)}\right] \qquad V = m_s \left[\frac{\psi'_n(m_s x_c)}{\psi_n(m_s x_c)}\right] - m_c \left[\frac{\psi'_n(m_c x_c)}{\psi_n(m_c x_c)}\right] \tag{7}$$

$$W = -i\left(\left[\frac{\psi_n(m_s x_c)}{\psi_n(m_s x_s)}\right][\zeta_n(m_s x_s)\psi_n(m_s x_c)] - [\zeta_n(m_s x_c)\psi_n(m_s x_c)]\right). \tag{8}$$

Here, we have expressed everything in terms of the dimensionless size parameter of the core and shell ($x_c$ and $x_s$ respectively) and the complex refractive indices of the core and the shell ($m_c$ and $m_s$ respectively). $\psi_n(z) = zj_n(z)$, where $j_n(z)$ is the spherical Bessel function of the first kind and $\zeta_n(z) = zh_n^{(1)}(z)$ where $h_n^{(1)}(z)$ is the spherical Hankel function of the first kind (Bohren and Huffman, 1983). In the event that $m_c = m_s$ the values of $U$ and $V$ both go to zero and equations 5 and 6 reduce to the solution for homogeneous spheres:

$$a_n = \left[\frac{\psi_n(x_s)}{\zeta_n(x_s)}\right] \frac{\left(\left[\frac{\psi'_n(m_s x_s)}{\psi_n(m_s x_s)}\right] - m_s \left[\frac{\psi'_n(x_s)}{\psi_n(x_s)}\right]\right)}{\left(\left[\frac{\psi'_n(m_s x_s)}{\psi_n(m_s x_s)}\right] - m_s \left[\frac{\zeta'_n(x_s)}{\zeta_n(x_s)}\right]\right)} \qquad b_n = \left[\frac{\psi_n(x_s)}{\zeta_n(x_s)}\right] \frac{\left(m_s \left[\frac{\psi'_n(m_s x_s)}{\psi_n(m_s x_s)}\right] - \left[\frac{\psi'_n(x_s)}{\psi_n(x_s)}\right]\right)}{\left(m_s \left[\frac{\psi'_n(m_s x_s)}{\psi_n(m_s x_s)}\right] - \left[\frac{\zeta'_n(x_s)}{\zeta_n(x_s)}\right]\right)} \tag{9}$$

Algorithmically, each of the six unique terms in square brackets in equations (5 - 8) is computed via its own recursion relation, which is critical to ensure stability (Toon and Ackerman, 1981). Note that the logarithmic derivative $\psi'_n(z)/\psi_n(z)$ must be calculated using downward recurrence in this (Wiscombe, 1979), but not all (Shiloah, 2018), formulations of the problem, while the other terms can be computed with upward recurrence. The recurrence relations are:

$$\frac{\psi'_{n-1}(z)}{\psi_{n-1}(z)} = \frac{n}{z} - \left(\frac{n}{z} + \frac{\psi'_n(z)}{\psi_n(z)}\right)^{-1}, \qquad \frac{\psi'_{n_{\max}+20}(z)}{\psi_{n_{\max}+20}(z)} = 0 \tag{10}$$

$$\frac{\zeta'_n(z)}{\zeta_n(z)} = \left(\frac{n}{z} - \frac{\zeta'_{n-1}(z)}{\zeta_{n-1}(z)}\right)^{-1} - \frac{n}{z}, \qquad \frac{\zeta'_0(z)}{\zeta_0(z)} = -i \tag{11}$$



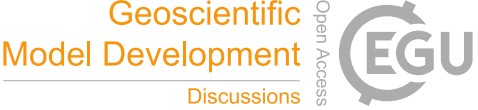

$$\frac{\psi_{n+1}(z)}{\zeta_{n+1}(z)} = \frac{\psi_n(z)}{\zeta_n(z)} \left( \frac{n+1}{z} - \frac{\psi_n'(z)}{\psi_n(z)} \right) \left( \frac{n+1}{z} - \frac{\zeta_n'(z)}{\zeta_n(z)} \right)^{-1}, \qquad \frac{\psi_1(z)}{\zeta_1(z)} = \frac{1}{2}e^{2iz}\frac{z+i}{z-i} + \frac{1}{2} \tag{12}$$

$$\zeta_n(z_1)\psi_n(z_2) = \frac{\zeta_{n-1}(z_1)\psi_{n-1}(z_2)}{\left( \frac{\zeta_n'(z_1)}{\zeta_n(z_1)} + \frac{n}{z_1} \right)\left( \frac{\psi_n'(z_2)}{\psi_n(z_2)} + \frac{n}{z_2} \right)}, \qquad \zeta_0(z_1)\psi_0(z_2) = \frac{1}{2}e^{-iz_1+iz_2} - \frac{1}{2}e^{-iz_1-iz_2} \tag{13}$$

$$\frac{\psi_n(z_1)}{\psi_n(z_2)} = \frac{\psi_{n-1}(z_1)}{\psi_{n-1}(z_2)}\frac{\frac{\psi_n'(z_2)}{\psi_n(z_2)} + \frac{n}{z_2}}{\frac{\psi_n'(z_1)}{\psi_n(z_1)} + \frac{n}{z_1}}, \qquad \frac{\psi_0(z_1)}{\psi_0(z_2)} = \frac{e^{-i(z_1+z_2)} - e^{i(z_1-z_2)}}{e^{-2iz_2} - 1} \tag{14}$$

where the square brackets have been dropped to avoid clutter. The variable $n_{\max}$ is the number of Mie coefficients necessary
to calculate to ensure convergence and is estimated as:

$$n_{\max} = \mathrm{Re}\{m_s\}x_s + 4.3(\mathrm{Re}\{m_s\}x_s)^{\frac{1}{3}} + 3 \tag{15}$$

based on Johnson (1996). The additional 20 terms calculated for the downward recurrence in Eq. 10 are discarded. Likewise, in
equations 1-4 the 0-th terms of $a_n$ and $b_n$ are not needed and so the 0th terms of the above recurrence relations (Eqs. 10-14) can
also be discarded. The TAMie package includes a separate, more efficient, subroutine for the homogeneous sphere case that
shares subroutines for calculating the recursion relations in Eqs. 10, 11 and 12 with the coated sphere code, and in the event
that the coated sphere function is called with $m_c = m_s$, $x_c/x_s < 0.01$, or $x_c/x_s > 0.99$ uses the homogeneous sphere solution
instead (using the refractive index of just the shell or just the core). This algorithm has been implemented in the "sphere"
and "coreshell" subroutines of the TAMie.py Python script released alongside this manuscript. TAMie has been implemented
to leverage numba (Lam et al., 2015) just-in-time compiling. Numba is a software library that compiles Python programs
directly to machine code. It has very limited impact on the structure of the Python script and only requires the addition of
function decorators to each of the compiled subroutines. Using this approach, TAMie achieves computation times comparable
to Fortran Mie codes while retaining the simplicity and readability of Python code.

While the above formulas have all been published by past authors, we have opted to reproduce them here in a concise manner
yet in enough detail that our Mie solver can be re-produced without external references, and to provide a description that uses
notation that is consistent with both our Python code and the remainder of this manuscript. For more detailed explanations of
the reasoning behind the algorithm's construction and the choice and stability of the recursion relations see Toon and Ackerman
(1981) and Wiscombe (1979).

## 2.2 Mie code testing

To ensure TAMie functions properly, we compared it with several established Mie scattering codes. We chose to test against
the Wiscombe (1993) Fortran implementation of the Toon and Ackerman (1981) algorithm for core-shell scattering, because it
proved less likely to crash than their original implementation. For homogeneous spheres, we tested against Fortran code from





| | $Q_e$ | | $Q_s$ | | $g$ | |
|---|---|---|---|---|---|---|
| Percentile | 90 | 99.9 | 90 | 99.9 | 90 | 99.9 |
| **Coated Spheres** | | | | | | |
| DMiLay vs TAMie | $9.2 \times 10^{-5}$ | 0.11 | 0.00013 | 0.12 | $2.9 \times 10^{-5}$ | 0.029 |
| PyMieScatt vs TAMie | 0.0068 | 0.099 | 0.0076 | 0.13 | 0.0021 | 0.036 |
| DMiLay vs PyMieScatt | 0.0056 | 0.077 | 0.0062 | 0.1 | 0.0016 | 0.03 |
| **Homogeneous Spheres** | | | | | | |
| BHMIE vs TAMie | $7 \times 10^{-5}$ | 0.00045 | $7 \times 10^{-5}$ | 0.00044 | $1.7 \times 10^{-5}$ | 0.00011 |
| PyMieScatt vs TAMie | $7.4 \times 10^{-5}$ | 0.00044 | $6.7 \times 10^{-5}$ | 0.00043 | 0.00058 | 0.00087 |
| BHMIE vs PyMieScatt | $1.3 \times 10^{-5}$ | $7.8 \times 10^{-5}$ | $6.9 \times 10^{-6}$ | $3.4 \times 10^{-5}$ | 0.00058 | 0.00087 |

**Table 1.** 90[th] and 99.9[th] percentile absolute differences between the outputs from TAMie and various other Mie scattering codes for both the coated and homogeneous sphere cases computed on a testing set of $10^6$ randomly generated particles.

| Code | Citation | Sphere run-time (s) | Core-shell run-time (s) |
|---|---|---|---|
| BHMIE | (Bohren and Huffman, 1983) | 5 | - |
| DMiLay | (Wiscombe, 1993) | - | 15 |
| TAMie | - | 6 | 15 |
| TAMie (no numba) | - | 22 | 138 |
| PyMieScatt | (Sumlin et al., 2018) | 156 | 431 |

**Table 2.** Time required for single CPU-core Mie calculations for all $10^6$ test cases (on a Ryzen 9 3900xt CPU).

Bohren and Huffman (1983). Finally, we performed a comparison to the PyMieScatt Python package (Sumlin et al., 2018) for both homogeneous and coated sphere cases. We tested the code on a dataset of 1M randomly generated layered particles, where the size parameter was randomly selected from $\ln\mathcal{U}(10^{-2}, 10^2)$ and the ratio of the core to shell radii was drawn from $\mathcal{U}(0.01, 0.99)$. The complex refractive indices of the core and shell were drawn independently from $m = \mathcal{U}(1.1, 3.0) + \ln\mathcal{U}(10^{-8}, 1)i$. Here, $\mathcal{U}$ and $\ln\mathcal{U}$ represent uniform and log-uniform distributions respectively, with the bounds shown in the parenthesis. The same dataset was used for both the layered and homogeneous sphere cases, and when the homogeneous sphere cases were tested the shell size parameter and refractive index were used to represent the whole particle. Table 1 shows the 99th, and 99.9th-percentile absolute differences between the various algorithms and Table 2 shows the single-core run-time required by each code to evaluate all 1M test particles.

Percentiles of the absolute differences over the testing dataset are shown in Table 1 instead of maximum differences because much more appreciable discrepancies can occur in the most extreme cases for the coated-sphere dataset. The largest absolute differences between the scattering models tend to occur in cases where both $\mathrm{Re}\{m_s\}$ and $x_s$ are large and many Mie coefficients must be calculated. The largest difference between DMiLay and TAMie occurred for such a particle, for which DMiLay output



$Q_e = 1.9$ and $Q_s = 2.6$ while TAMie output $Q_e = 2.1$ and $Q_s = 1.2$. Here TAMie's result is more physically plausible because $Q_s$ must be less than or equal to $Q_e$. In the worst case for PyMieScatt it output a completely non-physical scattering efficiency of 160. We investigated these large disagreements by plotting curves comparing each scattering code to TAMie for a range of $x_s$ and the same refractive indices and $x_c/x_s$ ratio that caused significant disagreement with TAMie in the testing dataset (Figure 1). These plots indicate that in both cases it was DMiLay and PyMieScatt generating the erroneous values, and spurious

spikes appear in their $Q_s$ curves for specific values of $x_s$. The most extreme differences between the Mie codes were negligible in the homogeneous sphere case, where BHMIE and TAMie disagreed by 0.053 on an extinction efficiency and PyMieScatt and TAMie disagreed by 0.045 also on an extinction efficiency.

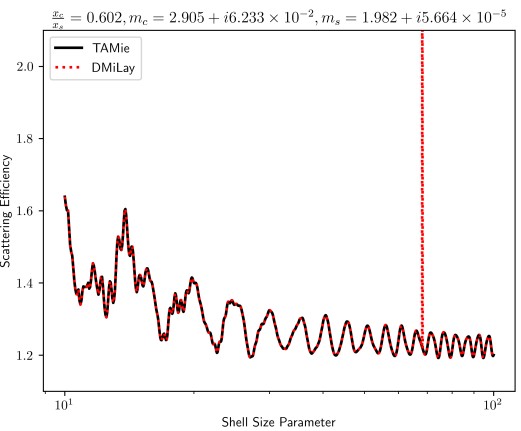
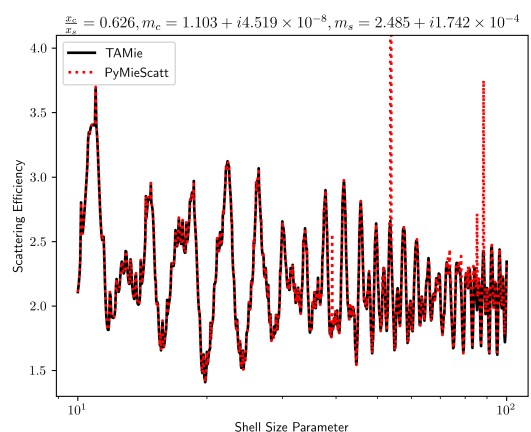

**Figure 1.** A comparison of scattering efficiency for a range of large size parameters in the specific test cases with the largest disagreement between TAMie and DMiLay (panel a) and TAMie and PyMieScatt (panel b).

     Finally, we performed some additional checks on TAMie for specific input limits with known behaviour on smaller ($10^5$ samples) testing sets. For particles with size parameters sampled from $\ln\mathcal{U}(10^{-4}, 10^5)$, the output from the core-shell model

with $m_s = m_c$ never diverged from the spherical scattering model's output by more than $10^{-14}$ (disabling the core-shell code's ability to call the homogeneous sphere code when $m_s = m_c$). In the Rayleigh scattering limit ($x << 1$), it is appropriate to use the approximation (Bohren and Huffman, 1983):

$$Q_e = Q_s + Q_a \qquad Q_s = \frac{8}{3}x^4 \left| \frac{m^2-1}{m^2+2} \right|^2 \qquad Q_a = 4x\mathrm{Im}\left\{ \frac{m^2-1}{m^2+2} \right\} \left( 1 + \frac{4x^3}{3}\mathrm{Im}\left\{ \frac{m^2-1}{m^2+2} \right\} \right). \tag{16}$$

For random inputs with $x_s$ drawn from $\ln\mathcal{U}(10^{-5}, 10^{-2})$ the maximum absolute difference between the efficiencies output from

TAMie and this approximation was $4.2 \times 10^{-5}$. Finally, in the geometric limit ($x >> 1$) the extinction efficiency approaches 2. Even for particles with $x_s > 100$ the solution may still oscillate around 2 as $x_s$ changes however. For a testing set of random inputs with $x_s$ sampled from $\ln\mathcal{U}(10^2, 10^5)$, the average $Q_e$ was 2.02 with a standard deviation of 0.04.





In closing, TAMie provides a new, entirely Python based, code for computing the optical properties of homogeneous and coated spheres. It shows good agreement with existing Fortran and Python Mie scattering codes and appropriate behaviour in the Rayleigh and Geometric limits, and when the core-shell solution approaches the homogeneous sphere solution. In the cases where there were large disagreements between TAMie and existing codes it appears that the discrepancies originated with the existing algorithms. In the following sections we use outputs from TAMie to train a neural network emulator to predict the bulk optical properties of aerosol populations. We hope though, that TAMie will prove useful to future investigators who wish to perform scattering calculations in Python.

## 3 Bulk optics

Mie codes can accurately calculate the optical properties of individual particles. Real atmospheric aerosol populations are not mono-disperse however. To calculate the bulk optics of a particle distribution with Mie code, one must compute optical properties for all relevant particle sizes and then integrate them over the particle size distribution. The repeated calls to Mie code necessary to do this, alongside the fact that the procedure must be repeated for different wavelengths and aerosol species, make this approach computationally infeasible in global atmosphere simulations. Instead, bulk AOPs are parameterized.

### 3.1 Bulk aerosol optics calculations

Bulk optics parameterizations typically estimate three quantities for an aerosol population: the dimensionless bulk asymmetry parameter ($\overline{g}$) and the mass absorption and scattering coefficients, $k_a$ and $k_s$ respectively, which have units of (m$^2$ kg$^{-1}$). Alternatively, the mass extinction coefficient ($k_e$) and/or the dimensionless single scattering albedo ($\omega$) might be used. These parameters satisfy the constraints: $k_e = k_s + k_a$ and $\omega = k_s k_e^{-1}$ and are thus interchangeable. The mass extinction coefficient for a poly-disperse aerosol population is defined:

$$k_e = \frac{\int_0^\infty p(r) Q_e(r, \lambda, m) \pi r^2 dr}{\int_0^\infty p(r) \rho \frac{4\pi}{3} r^3 dr} \tag{17}$$

where the subscript $e$ can be interchanged with $a$ or $s$ to define the mass absorption or scattering coefficients respectively (Petty, 2006). Here, $p$ represents the particle size distribution and $\rho$ is the density of the aerosol species. The mass extinction coefficient is defined in such a way that multiplying by the aerosol mass mixing ratio $M$ (kg kg$^{-1}$), air density $\rho_a$ (kg m$^{-3}$), and path length $\Delta z$ (m) yields the aerosol optical depth $\tau$:

$$\tau = k_e \rho_a M \Delta z. \tag{18}$$

The bulk asymmetry parameter $\overline{g}$ is defined in a similar way:

$$\overline{g} = \frac{\int_0^\infty g(r) p(r) Q_s(r, \lambda, m) r^2 dr}{\int_0^\infty p(r) Q_s(r, \lambda, m) r^2 dr} \tag{19}$$





(Petty, 2006), and can be thought of as a weighted average of the particle asymmetry parameters. Finally, modal aerosol models assume a different log-normal size distribution for each of their aerosol modes of the form:

$$p(r) = \ln \mathcal{N}(r, \mu, \sigma) = \frac{1}{r \ln(\sigma) \sqrt{2\pi}} e^{-\left(\frac{\ln(r/\mu)^2}{2 \ln(\sigma)^2}\right)} \tag{20}$$

where $\mu$ and $\sigma$ represent the geometric mean and standard deviation of the particle size distribution, $r$ is the particle radius, and $\ln \mathcal{N}$ represents a log-normal distribution.

## 3.2    E3SM parameterization

NeuralMie is meant to be model agnostic, but our intended use is deployment in the E3SM (Golaz et al., 2019) Atmosphere Model version 1 (EAMv1) (Rasch et al., 2019). EAMv1 simulates aerosols with the 4-mode version of the Modal Aerosol Module (MAM4) (Liu et al., 2012, 2016; Wang et al., 2020), which splits aerosol populations between 4 different size modes: Aitken, Accumulation, Coarse, and Primary Carbon, each with an assumed log-normal size distribution. When EAM's radiation
code is run (RRTMG (Iacono et al., 2008; Pincus and Stevens, 2013)), it calls a parameterization (Ghan and Zaveri, 2007; Ghan et al., 2001) that estimates the bulk optics of the aerosols represented by MAM4. We note that this parameterization is also used in other ESMs, including the Community Earth System Model v2.2 (Danabasoglu et al., 2020).

The existing optics parameterization uses a lookup table based approach to estimate bulk AOPs based on values that were pre-computed using Mie code. The lookup table is 5-dimensional and takes information about the wavelength (32), aerosol
mode (4), geometric mean radius of the size distribution (5), and the real (7) and imaginary (10) components of the refractive index as input, where the values in the parenthesis represent the table's resolution along the corresponding dimension. The high dimensionality of the table means that its resolution cannot be increased without making it significantly larger. When called, the parameterization estimates AOPs by performing 2-D interpolation with respect to the components of the refractive index and Chebyshev (Vetterling et al., 1988) interpolation along the geometric mean radius dimension, while the other two dimensions
are discrete and do not require interpolation. The Chebyshev interpolation along the geometric mean radius dimension allows this dimension to be stored at lower resolution than would otherwise be required. Ultimately, 3 such tables are required to store mass scattering coefficients, mass absorption coefficients, and the bulk asymmetry parameter.

The coarse resolution of the lookup tables used in E3SM's current parameterization introduce error in simulated AOPs. Ghan and Zaveri (2007) do not report errors for a large testing dataset, but evaluate several important test cases and report
typical errors less than 20%. Geiss et al. (2023) performed a more in-depth comparison of an ML emulator to EAM's existing parameterization and found that using neural networks for the same task could reduce error by multiple orders of magnitude, to the point where it is negligibly small, and other sources of error like sphericity and mixing assumptions likely dominate.

## 3.3    An ML-friendly formulation

Mie calculations depend only on size parameter ($x = \frac{2\pi r}{\lambda}$) and index of refraction, meaning that two particles with the same
refractive index but different sizes can have identical scattering properties at the appropriate wavelengths. This is a symmetry that can be leveraged to reduce the dimensionality of the optics parameterization problem. The only difficulty is that the





parameterization predicts bulk AOPs integrated over a particle size distribution, and to leverage this property we must re-formulate the bulk problem in terms of the size parameter. While a neural network can certainly be trained to predict $k_e$ as a function of both $\mu$ and $\lambda$, a goal of this study is to develop a neural network emulator that is significantly more generalizable and

flexible than in Geiss et al. (2023). Re-formulation in terms of the size parameter means that the trained network will be able to generalize to wavelengths and particle size ranges that fall outside of its training dataset, a rarity in typical neural network applications (Haley and Soloway, 1992; Xu et al., 2020). Furthermore, ML models can often benefit from dimensionality reduction on their input space (Murphy, 2012) and here that can be done by leveraging a known symmetry in the underlying physical system. To this end, we combine Equations 17 and 20 and introduce the following substitutions:

$$r = \frac{\lambda x}{2\pi} \qquad \mu = \frac{\lambda \mu_x}{2\pi} \qquad \frac{dr}{dx} = \frac{\lambda}{2\pi} \tag{21}$$

$$k_e \lambda \rho = \frac{3\pi \int_0^\infty e^{-\left(\frac{\log(x/\mu_x)^2}{2\ln(\sigma)^2}\right)} Q_e(x,m) x \, dx}{2 \int_0^\infty e^{-\left(\frac{\log(x/\mu_x)^2}{2\ln(\sigma)^2}\right)} x^2 \, dx}. \tag{22}$$

The scaling factor in front of the exponential in $p(r)$ and most of the constants introduced by the substitutions cancel out. This leaves only one instance of $\lambda$ and $\rho$ which can be pulled out of the integral and moved to the left hand side. The value of $\lambda\rho$ will be known at inference time so we opt to train our neural network to predict the function on the right-hand side in

Equation 22 and can simply divide its predictions by $\lambda\rho$. With this formulation, the neural network requires $\mu_x$ as an input rather than both $\mu$ *and* $\lambda$, and the input dimensionality has been reduced by 1.

We have designed NeuralMie to predict 3 output values: $(k_e \lambda \rho)$, $\omega$, and $\overline{g}$. $\omega$ and $\overline{g}$ are dimensionless parameters, and it can be seen by inspection of Eqs. 19 and 22 and from the relation $\omega = k_s k_e^{-1}$ that performing the substitutions in Eq. 21 results in all of the extra terms introduced canceling and neither $\omega$ or $\overline{g}$ need to be scaled to be predicted as functions of $\mu_x$. For $k_e$,

scaling by $\lambda\rho$ allows it to be predicted as a function of $\mu_x$ and conveniently yields a non-dimensional value, though values of $k_e \lambda \rho$ still span multiple orders of magnitude in practice.

Re-formulating the bulk optics problem in this way is beneficial from an ML perspective for 3 main reasons: 1) the range of possible values for $k_e \lambda \rho$ is smaller than $k_e$ because of the wavelength scaling. Meanwhile the other two predictands remain bounded to the range [0,1] (mathematically, $\overline{g}$ is in the range [-1,1], but for the aerosol populations considered here is non-

negative), which is convenient for neural networks because a sigmoid output activation can be used for these two values. 2) the dimensionality of the input space has been reduced by one, making this a simpler problem to solve. 3) a neural network trained to perform predictions in terms of $\mu_x$ can extrapolate and make predictions for values of $\mu$ and $\lambda$ that were not in the range of the training data as long as the associated value of $\mu_x$ was.





### 3.4 Bulk optics in the Rayleigh limit

In the Rayleigh limit it is not necessary to use a neural network emulator to calculate bulk AOPs. Eq. 17 with Eqs. 16 and 20 plugged in for $Q$ and $p$, respectively, has an analytical solution in terms of the error function:

$$k_s = \frac{4\pi\mu_x^3}{\lambda\rho}e^{\frac{27}{2}\ln(\sigma)^2}\left|\frac{m^2-1}{m^2+2}\right|^2 \frac{\left[\text{erf}\left\{\frac{6\ln(\sigma)^2-\ln(x/\mu_x)}{\sqrt{2}\ln(\sigma)}\right\}\right]_{x_1}^{x_2}}{\left[\text{erf}\left\{\frac{3\ln(\sigma)^2-\ln(x/\mu_x)}{\sqrt{2}\ln(\sigma)}\right\}\right]_{x_1}^{x_2}} \qquad k_a = \frac{6\pi}{\lambda\rho}\text{Im}\left\{\frac{m^2-1}{m^2+2}\right\} \tag{23}$$

where we have dropped the $x^4$ term in the approximation of $Q_a$, since the $x$ term will dominate for small values of $x$. For the core-shell case we simply use volume-weighted mixing of the refractive index in this limit. We also assume $g \to 0$. The use

of the Rayleigh approximation in this limit allows the neural network to focus only on learning cases where Mie scattering is relevant. This approximation was appropriate to use for about 22% of the testing data generated for NeuralMie (discussed in Section 4.2). The mean absolute errors on that data for $\omega$ and $\overline{g}$ were 0.0002 and 0.0017 respectively. In Section 4 we use the mean absolute percentage error (MAPE) to evaluate estimated mass extinction coefficients, but in this limit many of the values of $k_e$ are very close to zero and the formulation for MAPE means it can explode for very small inputs. Meanwhile, a metric

like mean absolute error (MAE) will be completely dominated by the handful of very large values of $k_e$. As a compromise, we report that using the approximation in Eq. 23, the MAPE when the true value of $k_e\lambda\rho$ is 0.01 or greater is $0.048\%$ while the MAE when the true value of $k_e\lambda\rho$ is less than 0.01 is $1.86 \times 10^{-6}$.

In the next Section we introduce NeuralMie, which is meant to be used alongside this approximation. In the event that the inputs can be accurately computed using the Rayleigh approximation, Eq. 23 is significantly faster to evaluate than performing

inference with the neural network, and using this approach reduces the range of behaviours the neural network needs to learn.

## 4 NeuralMie

This section introduces "NeuralMie," a neural network based emulator for bulk AOPs. It consists of two different neural networks, one trained to represent scattering by log-normally distributed populations of homogeneous spherical particles and one for coated spherical particles. The model has been created with use in E3SM in mind, though is designed in such a way

that it should be compatible with other Earth system models, climate models, and weather models, and can be used as a general purpose and fast alternative to Mie code for particle populations with refractive indices in the range of values it was trained on (Table 5).

### 4.1 Setup

Both the homogeneous and coated sphere cases use a four-layer feed forward neural network. All of the model's internal

connections use the "swish" activation with $\beta = 1$ (Chollet et al., 2015) (sometimes called the "SiLU" or "sigmoid linear unit" activation function (Ramachandran et al., 2017; Hendrycks and Gimpel, 2016)). In the homogeneous sphere case, the neural network takes 4 inputs: $\mu_x = 2\pi\mu/\lambda$, $\sigma$, $\text{Re}\{m\}$, and $\text{Im}\{m\}$ (in that order). These are the geometric mean radius of



| Variable: | $\mu_x = 2\pi\mu/\lambda$ | $\sigma$ | $Re\{m\}$ | $Im\{m\}$ | $f = x_c/x_s$ |
|---|---|---|---|---|---|
| Scaling function: | $(\ln(\mu_x) + 1.5)/3.6$ | $2\sigma - 4$ | $2Re\{m\} - 4$ | $(\ln(Im\{m\}) + 9)/5$ | $4f - 2$ |

**Table 3.** Neural network input scaling functions. These map from physical input variables to dimensionless inputs that are approximately uniformly distributed in the range -2 to 2.

| Variable: | $k_e$ | $\omega$ | $g$ |
|---|---|---|---|
| Scaling function: | $e^{o_1}/(\lambda\rho)$ | $1/(1 + e^{-o_2})$ | $1/(1 + e^{-o_3})$ |

**Table 4.** Neural network output scaling functions. These map from the three dimensionless outputs from NeuralMie's output layer (denoted $o_1$, $o_2$, and $o_3$ above) to physical values of $k_e$, $\omega$, and $g$.

the size distribution expressed as a size parameter, the geometric standard deviation of the size distribution, and the real and imaginary components of the aerosol's refractive index respectively. In the core-shell case, it takes 7 inputs, two additional values representing the core's refractive index, and a value $f = x_c/x_s$ representing the ratio of the core's radius to the particle radius. Each of the inputs is scaled to a range of approximately -2 to 2 using the transforms shown in Table 3.

The model's output layer does not apply an activation function and outputs three values $o_1$, $o_2$, and $o_3$, corresponding to $k_e$, $\omega$, and $\overline{g}$. To convert the outputs to physical values the transforms in Table 4 must be applied. The choice to use linear outputs was made largely because of the limitations of the Fortran Keras bridge (Ott et al., 2020), which we are using to deploy the model in Fortran.

Outputting $k_e$ scaled by the wavelength and density is critical for ensuring the flexibility of NeuralMie. Firstly, the density is not necessary for scattering calculations and this value can be applied after inference. More importantly, scaling by the wavelength allows for the neural network to represent scattering entirely in terms of the dimensionless size parameter which makes its performance largely independent of wavelength and particle size (geometric mean radius). We opted to predict the single scattering albedo in addition to $k_e$ and $\overline{g}$ because it is bounded by 0 and 1. This means we can apply a sigmoid function to the neural network's output to bound it to a physical range. Constructing the outputs this way can be thought of as predicting the mass extinction coefficient and then predicting what fraction of it is partitioned into scattering versus absorption. This is an implicitly enforced analytical constraint (Beucler et al., 2021), albeit a simple one, in that the network's output cannot break the constraints $k_e = k_a + k_s$ and $\omega = k_s k_e^{-1}$ (Bohren and Huffman, 1983).

### 4.2 Datasets

The training, validation, and testing datasets were generated by randomly sampling a large range of plausible values of the various input parameters. The ranges sampled for each input variable have been chosen to span the range possible inputs that could be produced by the various versions of MAM. The range and distribution of values sampled for each variable are given in Table 5.





| Variable: | $\lambda(m)$ | $\mu(m)$ | $\sigma$ | $Re\{m\}$ | $Im\{m\}$ | $f = r_c/r_s$ |
|---|---|---|---|---|---|---|
| Distribution: | $\ln\mathcal{U}$ | $\ln\mathcal{U}$ | $\mathcal{U}$ | $\mathcal{U}$ | $\ln\mathcal{U}$ | $\mathcal{U}$ |
| Range: | $(2 \times 10^{-7}, 10^{-3})$ | $(5 \times 10^{-9}, 5 \times 10^{-5})$ | $(1.2, 2.8)$ | $(1.1, 3.0)$ | $(10^{-8}, 1)$ | $(0, .98)$ |

**Table 5.** Random distributions and sampling ranges used to generate the training, testing, and validation inputs.

To train and evaluate NeuralMie, we generated a dataset of 100M randomly generated log-normal aerosol size distributions. Of these, we allocated the first 80M as training data, the subsequent 10M as validation data, and the final 10M as testing data. Because each sample was generated randomly and independently, the validation and testing data are uncorrelated with the training data, but may contain individual samples that are nearby specific points in the training dataset. In order to ensure the model has not overfit within its support, it will be important to analyze not just the mean error but the spread of errors for the

testing set when evaluating the model (Section 4.5).

Each of the input samples has all 7 inputs required by the core-shell model. We generated two sets of target data though, one for core-shell scattering and one for the homogeneous sphere case. In the homogeneous sphere case, only the first input refractive index ($m_s$) is used for the whole particle and $f$ is discarded. The target data were calculated by numerical integration of Eq. 17 in log-coordinates, spanning 1024 logarithmically spaced values of $r$. Extinction, scattering, and absorption efficiencies

for each value of $r$ were computed using TAMie. The integration bounds were determined using the log-normal cumulative density function to encompass 99.9% of the area under the size distribution.

As discussed in Section 3.4, in some cases it is sufficient to use a Rayleigh scattering approximation to compute the bulk optics, and the randomly generated inputs included some of these cases. We chose to retain those cases for testing of our Rayleigh approximation function, but did not include them in training or evaluation of the ANNs because inputs in this regime

will not be fed to the ANN in deployment. This reduced the size of each of the training, validation, and testing sets by around 22%. We opted to use the Rayleigh approximation (Eq. 23) when the upper bound of integration was less than or equal to $x = 0.1$. Specifically, the criteria is computed:

$$\mu e^{\sqrt{2}\ln(\sigma)\mathrm{erf}^{-1}(0.999)} \leq 0.1 \tag{24}$$

where $\mathrm{erf}^{-1}$ is the inverse error function.

## 4.3 Training procedure

The neural networks were trained in two stages. In the first stage, we performed a limited hyperparameter search that involved relatively short training runs to determine optimal model sizes for each of the two ANNs. During this stage, models were trained on the training dataset and evaluated on the validation dataset. In the second stage, once model architectures were





selected, a final training run was performed on both the training and validation data and these models were evaluated on the
testing dataset.

In both phases, we used the Adam optimizer (Kingma and Ba, 2014) with an initial learning rate of 0.001. We applied
learning rate reductions by a factor of 10 at 30, 60, and 90% of the way through training. We experimented with a range of
batch sizes and found that the model could train well with a large batch size (1000s of samples) but performed slightly better
with a much smaller batch size (tens of samples). Training on large batches is attractive because it allows for significant GPU
acceleration, so we opted to use a batch size of 2048 and train for 100 epochs during the hyperparameter tuning phase (about
1.5hrs per ANN on GPU) and a batch size of 64 and train for 33 epochs during the final training phase (8-12hrs per ANN on
CPU). These training durations and learning rate schedules were decided by manually monitoring the training loss and making
a qualitative decision as to when the model skill had stopped increasing appreciably. We found it useful to use $100\times$ the
hyperbolic tangent of the absolute fractional error for $k_e$ as a performance metric during training. This is similar to the mean
arc-tangent absolute percentage error, and approximates the MAPE with contributions from cases with extreme percentage
errors suppressed to 100%.

NeuralMie's outputs have different physical meaning and scaling, so we had to implement a custom loss function that treats
$k_e$ differently than $\omega$, and $\overline{g}$ (and performs the sigmoid scaling necessary to retrieve $\omega$ and $\overline{g}$ from the model's output layer).
Ultimately we decided to use the mean absolute error (MAE) for $\omega$ and $\overline{g}$ and use the root mean squared log error (RMSLE)
for $k_e\lambda\rho$. We found that simply summing the three losses was effective and a scaling parameter was not needed to combine
them. Specifically, the loss was computed as:

$$\mathcal{L} = \sqrt{\frac{1}{N}\sum_N (\ln(k_e\lambda\rho) - \widehat{\ln(k_e\lambda\rho)})^2} + \frac{1}{N}\sum_N (|\omega - \widehat{\omega}| + |g - \widehat{g}|) \tag{25}$$

where values with hats are predicted values while those without are target values and $\frac{1}{N}\sum_N$ represents an average with
respect to the samples in a training batch of size $N$. Recall that the neural networks do not apply an activation to their last layer,
so expressed in terms of the model's outputs, the loss is:

$$\mathcal{L} = \sqrt{\frac{1}{N}\sum_N (\ln(k_e\lambda\rho) - \widehat{o_1})^2} + \frac{1}{N}\sum_N (|\omega - \text{sig}(\widehat{o_2})| + |\overline{g} - \text{sig}(\widehat{o_3})|) \tag{26}$$

where "sig" represents the sigmoid function. The RMSLE can be computed using a ratio of the ground truth to the output,
so, while we have written them out here, values of $\lambda\rho$ do not need to be used in the loss function implementation.

## 4.4  Model selection

In our previous work (Geiss et al., 2023) we utilized a neural architecture search strategy (Elsken et al., 2019; Hutter et al., 2019)
that included random wiring of network layers (Xie et al., 2019), and found that this strategy produced more skilled models
than those found in a search of conventional, serially connected, architectures. This technique has been shown to be beneficial





in other applications in atmospheric science (Yik et al., 2023) likely due to its ability to identify specific skip connections (He et al., 2016) helpful for the target task. Ultimately we would like NeuralMie to be deployable in Fortran-based weather and

climate models however, and we are using the Fortran-Keras bridge to accomplish this which does not support complex model architectures (Ott et al., 2020). For this reason, we performed a more limited hyperparameter search of conventional serially connected ANNs to determine an optimal model. Future versions of E3SM will be written in C++ which will make deploying ML models significantly easier, and so we may eventually produce a superior version of NeuralMie leveraging a random wiring architecture search, but leave this as a future task for now.

Our parameter search and many of our model design choices are based on findings from Geiss et al. (2023). The main exception is that we have used "swish" transfer functions which we found to perform better than tanh. The Fortran Keras bridge (Ott et al., 2020) does not natively support swish, but it does support the sigmoid function and swish is simply the sigmoid times its input. We randomly generated 200 neural networks both for the homogeneous and coated sphere cases. The networks had randomly selected layer counts between 2 and 4 (Geiss et al., 2023) with equal numbers of neurons per layer

(along with a 3-neuron output layer) and randomly selected total trainable parameter counts between 500 and 100,000. They were trained as described in Section 4.3 and then evaluated on the validation set.

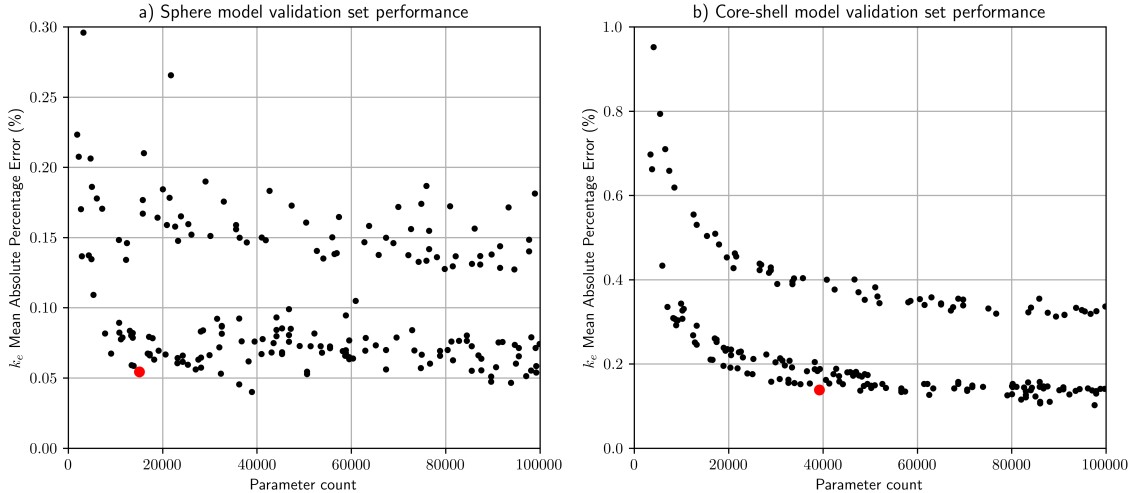

**Figure 2.** Mean absolute percentage error (MAPE) computed on the validation set for each of the randomly generated serially connected neural networks. Red dots represent the neural networks that were selected for use and provide a good balance between network size and performance.

Figure 2 shows the validation set MAPE for $k_e$ plotted against model complexity (in terms of number of parameters). In this plot we can clearly identify the Pareto frontier, and can strike a good balance between model accuracy and complexity by selecting a model for that lies near the elbow of the curve. Because this model is intended for use as a parameterization

that could easily be called millions of times during a climate or weather simulation, there is significant benefit to using as



few parameters as possible. Furthermore, accurate parsimonious models are less likely to overfit (Geman et al., 1992) and are generally more trustworthy as a result. The red markers in Figure 2 represent the architectures we eventually chose for additional training. The homogeneous sphere model has 69 neurons in each of four hidden layers while the core-shell model has 112 neurons in each of four hidden layers.

Both panels in Figure 2 contain two bands of models, where one set performs considerably worse than the others. The upper band in each case corresponds to models with only two hidden layers, indicating that including at least three hidden layers is critical, but moving beyond that depth does not result in significantly better performance. In our previous work (Geiss et al., 2023), we found that large networks with more than three hidden layers typically performed worse than those with three, which was not the case here. We hypothesize that this may be due to the use of the "swish" transfer function here instead of tanh.

The tanh function saturates for any large magnitude outputs from a layer and is thus more likely to suppress gradients when training deep ANNs (Bengio et al., 1994; He et al., 2016). More investigation would be needed to confirm this however.

### 4.5 Model performance

After selecting an model architecture for both the core-shell and homogeneous sphere problem, we performed a final round of training that included the training and validation data and used a much smaller batch size (Section 4.3). These final two models

were then evaluated on the testing set, and their testing errors for all output parameters are summarized in Figures 3 and 4 and in table 6. We chose to evaluate $k_e$ in terms of absolute percentage error because its possible values span multiple orders of magnitude. Also, because it is computed as a ratio we can compute this metric without accounting for the extra $\lambda\rho$ term in Eq. 22. Meanwhile, the asymmetry parameter and single scattering albedo are bounded by 0 and 1, so we have expressed their error in terms of absolute error.

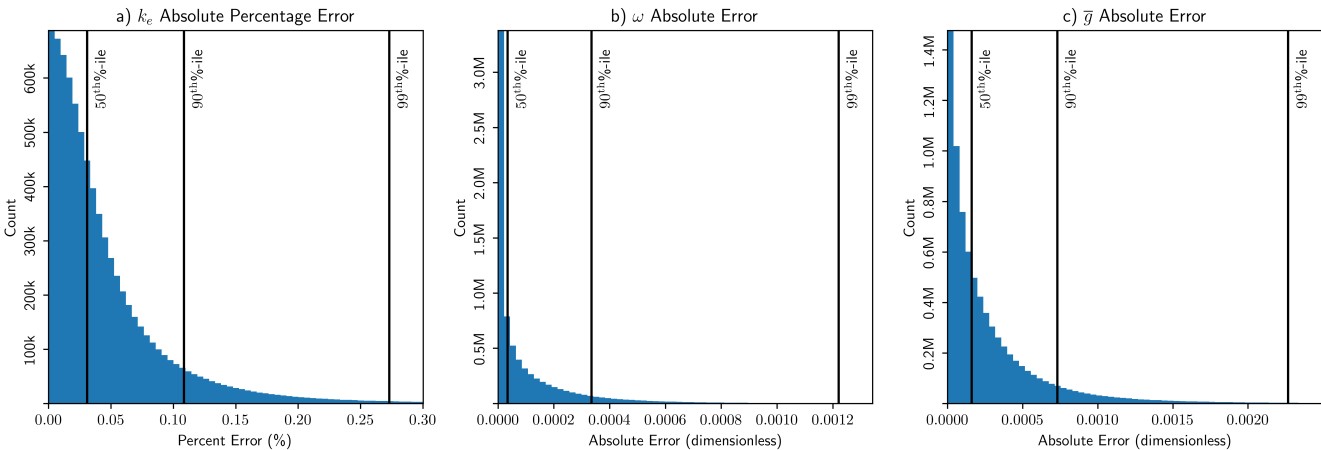

**Figure 3.** Error histograms for the homogeneous sphere Mie optics emulator.



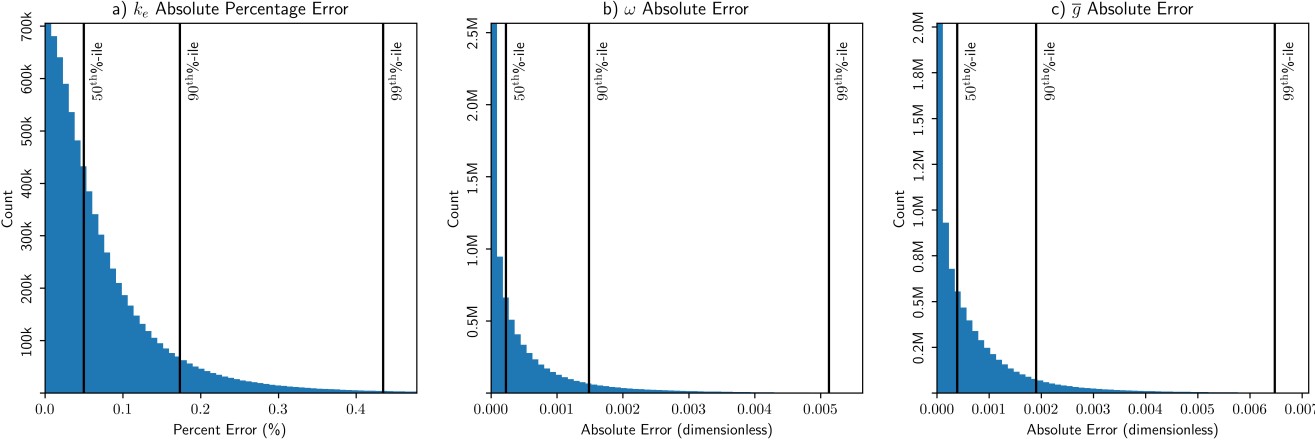

**Figure 4.** Error histograms for the coated sphere Mie optics emulator.

|  | Homogeneous Spheres | | | Coated Spheres | | |
| --- | --- | --- | --- | --- | --- | --- |
|  | $k_e$ (APE) | $\omega$ (AE) | $\overline{g}$ (AE) | $k_e$ (APE) | $\omega$ (AE) | $\overline{g}$ (AE) |
| Mean | 0.05% | $1.3 \times 10^{-4}$ | $3.1 \times 10^{-4}$ | 0.08% | $5.8 \times 10^{-4}$ | $8.1 \times 10^{-4}$ |
| 90th percentile | 0.11% | $3.3 \times 10^{-4}$ | $7.3 \times 10^{-4}$ | 0.17% | $1.5 \times 10^{-3}$ | $1.9 \times 10^{-3}$ |
| 99th percentile | 0.27% | $1.2 \times 10^{-3}$ | $2.3 \times 10^{-3}$ | 0.43% | $5.1 \times 10^{-3}$ | $6.5 \times 10^{-3}$ |
| 99.99th percentile | 0.66% | $6.9 \times 10^{-3}$ | $1.1 \times 10^{-2}$ | 1.86% | $2.4 \times 10^{-2}$ | $3.3 \times 10^{-2}$ |
| Maximum | 1.36% | $2.3 \times 10^{-2}$ | $3.7 \times 10^{-2}$ | 31% | 0.13 | 0.09 |

**Table 6.** NeuralMie errors on the testing set. Errors for the mass extinction coefficient ($k_e$) are expressed as Absolute Percentage Error (APE) because values of $k_e$ span multiple orders of magnitude, while errors for the single scattering albedo ($\omega$) and the bulk asymmetry parameter ($\overline{g}$) are expressed as Absolute Error (AE) because their values are constrained to a range of (0,1). The mean test set error, 90th percentile, 99th percentile, 99.99th percentile, and maximum error are shown to give an idea of the spread of the error. The maximum error for $k_e$ for the core-shell case is an extreme outlier and only 9 of the 10M test cases had error exceeding 10%.

Both models are highly performant, with negligible error compared to other sources of uncertainty affecting aerosol optics (e.g. aerosol burden, composition, size distribution, etc.). Both achieve mean absolute percentage errors on the order of hundredths of a percent, which is dramatically better than the conventional parameterization (Ghan and Zaveri, 2007; Geiss et al., 2023). Table 6 also shows several percentile errors and the maximum error for the test set. The maximum absolute percentage error for the homogeneous sphere case is only 1.36%. Meanwhile, the core-shell model had very low error even at the 99.99th

percentile of only 1.86%, but has a high maximum error of 31%. This is an extreme outlier case, and only 9 samples from the 10M sample testing set had $k_e$ errors exceeding 10%. These were typically particles with very high or very low values of $f$, so it may be preferable to simply use a homogeneous mixing assumption in cases where the core is extremely small or the shell is extremely thin. Figures 3 and 4 summarize the same information as Table 6 visually, and provide some insight into the spread



of the test error. Finally, absolute errors for $\omega$ and $\overline{g}$ were also very small, with errors on the order of or $10^{-3}$ out to the 99th
percentile.

Overall, this level of accuracy is a significant improvement over the existing parameterization. These models are accurate enough that further improvements likely wont have a substantial impact on weather and climate simulations, and they are nearly as good as running Mie code directly in a atmosphere simulation. In Geiss et al. (2023), we used a formulation of the problem that closely followed Ghan and Zaveri's (2007) parameterization. Here, we have used a different formulation and removed
some simplifications made by that algorithm, so our results are not directly comparable, and will differ for reasons other than improved parameterization accuracy. That said, they report seeing errors on the order of 10% for the test cases they evaluated and the MAPE of 0.05% or 0.08% observed here for a significantly larger and more diverse test set is a substantial improvement. Finally, Figure 5 shows a plot of specific extinction and scattering and the asymmetry parameter for a hypothetical population of sulfate coated black carbon. This can be compared to the plots in Ghan and Zaveri (2007) where there is visible disagreement
between parameterized and true AOPs, whereas here there is no visually perceptible difference between the output from Mie code and from NeuralMie.

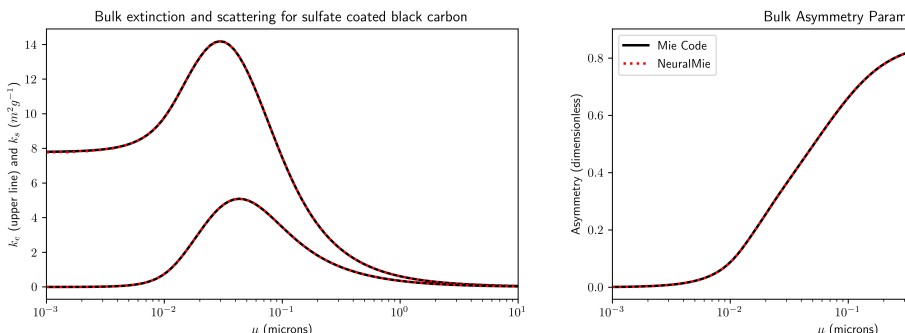

**Figure 5.** Scattering properties for log-normally distributed populations of sulfate coated black carbon with various values of $\mu$. The core has a refractive index of $m = 1.95 + i0.78$ and a density of $\rho = 1.8\text{g cm}^{-3}$ while the shell has a refractive index of $1.55 + i1 \times 10^{-8}$ and a density of $\rho = 1.2\text{g cm}^{-3}$, with $f = 0.25$. The upper lines in the left panel correspond to extinction while the lower line corresponds to absorption.

## 5 Discussion

Here we have presented to new contributions in the area of aerosol optics modeling: TAMie and NeuralMie. The new TAMie scattering code implements the Toon and Ackerman (1981) Mie scattering algorithm in Python and provides an easy to use
and easy to read Mie solver. It compares favorably to both established Fortran and Python Mie codes in terms of accuracy and stability, and achieves speeds comparable to Fortran solutions. It was extremely useful for producing the volume of data needed to train the NeuralMie optics emulator, and we sincerely hope that future investigators will find it useful as well. NeuralMie





provides a significant improvement in the accuracy with which aerosol optics are represented in E3SM. Not only does it add the capability to represent core-shell scattering, but it does so with an unusually high level of flexibility for a neural network

and incurs only a small fraction of the error of the parameterization that is currently deployed. In this section we discuss some of the limitations of the emulator, other approaches to aerosol optics emulation, and potential future use cases and areas of future research.

Neural Mie has several limitations that should be noted. The largest limitations are not necessarily specific to the machine learning model and are rooted in the simplifying assumptions about particle shapes and particle size distributions used in

aerosol models and used to create the training data. Our solution assumes that aerosol populations are always log-normally distributed. This is not true in reality but is a very common simplifying assumption in modal aerosol models. This means that the neural network model is only appropriate for use alongside these models or in specific cases where the log-normal assumption is reasonable. Similarly, the assumption of particle sphericity (and perfect concentric spheres in the core-shell case) deviates from reality. It is appropriate for some types of atmospheric particles but not others. Notably, atmospheric black

carbon can form highly irregular shapes that have significantly different scattering properties than spheres, though when it becomes coated in a large quantity of sulfate the sphericity assumption becomes more reasonable (Adachi et al., 2010). These differences in shape can substantially impact scattering properties. Finally, neural networks are typically poor extrapolators (Haley and Soloway, 1992; Xu et al., 2020). While we have designed this model to support nearly any wavelength and particle size combination, it should only be trusted when the input refractive indices and size distribution standard deviations are within

the range the model was trained on (i.e., Table 5 in this study).

This model was designed to directly estimate the bulk optics of a particle size distribution, but there are other approaches to the same problem that we briefly discuss here. One strategy is to train a neural network to emulate Mie code, and compute individual particle optical properties rather than compute bulk optics directly. Two past studies we are aware of have done this (Thong and Yoon, 2022; Stremme, 2019). A major benefit of this approach is that the resulting ML model will be applicable to

aerosol models that do not prescribe a particle size distribution (bin models for example), making it even more flexible. There are two main downsides to this method though. Firstly, to numerically compute the integral in Eq. 17, the ML model would need to be called multiple times to compute values of $Q_e$. This negates a significant amount of the performance improvement over Mie code, particularly considering that the number of Mie coefficients required scales with the size parameter, and in cases with a small size parameter a single call to Mie code can actually be much more efficient than even a small neural

network. Secondly, individual particles' optical properties vary much more rapidly as a function of size parameter than a log-normal aerosol populations' bulk AOPs vary as a function of the geometric mean radius (e.g. Figures 1 and 5). This is because integrating AOPs over a size distribution smooths out the variability in the individual particle optical properties. This makes the bulk AOPs much easier to predict than particle optics with a small neural network and is likely a large contributor to NeuralMie's extremely low error. Another, quite different, approach is to skip the process of estimating AOPs altogether. This

could be done in cases where the entire radiative transfer scheme is emulated, and it is conceivable that a radiative transfer emulator could be developed that ingests information about aerosol populations directly. This approach bypasses the need for a standalone optics parameterization because those calculations would be handled internally by the emulator. The downside



is that the emulator would have to be retrained to accommodate any changes to the aerosol model, and re-training such an emulator is much more difficult than retraining an aerosol optics emulator. We believe Neural Mie strikes a good balance
between generalizability, speed, and accuracy compared to these other approaches.

Work to integrate Neural Mie into E3SM is ongoing. Based on offline results, we expect that online performance will be comparable to running Mie scattering code directly in the model. Other investigators have found that integration of ML-based parameterizations into climate models can be tricky however, and even models that are very accurate offline can lead to instability when used online. Additionally, integration of the core-shell version of NeuralMie will require some additional
model development because MAM4 and E3SM are not currently designed to represent core-shell particles. Model integration and evaluation of the impact that the new emulator and core-shell optics have on simulated aerosol radiative effects will be an important area of future work. There are several other areas of potential future research and applications for NeuralMie. Firstly, black carbon has a particularly large radiative impact, and is the strongest absorbing common aerosol species (Bond et al., 2013). When mixed with other aerosol species the difference in absorption estimates under homogeneous versus core-
shell mixing assumptions can be substantial (Adachi et al., 2010), and NeuralMie adds the capability to represent coated black carbon particles which will improve the physical realism of E3SM. Black carbon particles can have highly irregular shapes however, which impact their optical properties (Adachi et al., 2010; Fierce et al., 2020), and investigation of methods to correct for the errors associated with NeuralMies's sphericity assumption for black carbon specifically would be valuable. Finally, our previous work (Geiss et al., 2023) leveraged a neural architecture search that included random wiring of network layers. We
did not do this here because of the limitations of the Fortran Keras bridge, but future versions of E3SM will be written in C++ which will make deployment of complicated neural networks significantly easier. In the future it may be worthwhile to further tune the NeuralMie architecture using this method to achieve even better accuracy. Ultimately, we expect that the increase in accuracy and capability introduced by NeuralMie will improve E3SM's ability to represent aerosol direct radiative effects.

*Code and data availability.*   The code and data associated with this manuscript are undergoing internal review at Pacific Northwest National
Laboratory before open source release. For now, we have provided access to the editor and reviewers via Zenodo and will make the code publicly available as soon as the internal review is finished.

*Author contributions.*   AG designed code and methods, performed experiments, wrote manuscript. PM conceived project, secured funding, edited manuscript

*Competing interests.*   PM is a member of the editorial board of Geoscientific Model Development. AG declares no competing interests.





*Acknowledgements.* This study was supported as part of the "Enabling Aerosol–cloud interactions at GLobal convection-permitting scalES (EAGLES)" project (project no. 74358), sponsored by the U.S. Department of Energy, Office of Science, Office of Biological and Environmental Research, Earth System Model Development (ESMD) program area. The Pacific Northwest National Laboratory is operated for DOE by Battelle Memorial Institute under contract DE-AC05-76RL01830.

The research used high-performance computing resources from PNNL Research Computing as well as resources from the National Energy Research Scientific Computing Center (NERSC), a U.S. Department of Energy Office of Science User Facility located at Lawrence Berkeley National Laboratory (operated under contract no. DE-AC02-05CH11231 using NERSC award nos. ALCC-ERCAP0016315, BER-ERCAP0015329, BER-ERCAP0018473, and BER-ERCAP0020990).

We thank Balwinder Singh and Sungduk Yu for their help with the Fortran Keras Bridge and assistance with E3SM integration. We also thank Laura Fierce, Rahul Zaveri, and Payton Beeler for their input regarding the optics of black carbon and sulfate-coated black carbon.





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
