# Peer review of "NeuralMie (v1.0): An Aerosol Optics Emulator"

_Geoscientific Model Development, 2024_

## Referee Comment (RC2)

**Review of manuscript gmd-2024-30**

June, 22, 2024

**Title**: NeuralMie (v1.0): An Aerosol Optics Emulator.
**Reference**: gmd-2024-30

**General review**

**Topic:** The manuscript describes the development of NeuralMie tool, an aerosol optics emulator based on machine learning methods, specifically neural networks that uses TensorFlow as its main external library. The manuscript also discusses the future implementation as part of E3SM, but the applications of the presented tool are not limited to the original one. The reviewers have had access to the code and data used to create this tool.

**Overview:** The manuscript content fits well within the scope of the journal Geoscientific Model Developments, and the tool described may be beneficial for other projects in the context of climate and atmospheric modelling.

**Evaluation:** The manuscript is well written: the scientific questions are properly explained and the model/method is reasonable. The main core of the article, which I understand is the NeuralMie tool, is interesting, and potentially useful (beyond the initial motivation of the authors are part of E3SM), for the community of scientists that implements optical properties of aerosols in climate models and for this reason I recommend **to accept the paper with minor comments** that I will explain in the review.

**1   Main review**

Here it is included a short evaluation of general points of the research presented:

**Key results** In my opinion, the key result is the NeuralMie tool and its development, while the TAMie is more a step of the development than a novelty in this research field.

**Validity** The full process of developing and implementing the tool is provided in sufficient detail to meet strict reproducibility requirements. The code/datasets are provided to reviewers and along the manuscript new developed tools compared well with established/checked tools. Therefore, it is expected that results are valid. Note however that the code is not provided with any help to reviewers to be executed (README or make files).

**Originality** The originality of the research is acceptable. There are earlier *emulators*, but the authors adequately justify the advantages of NeuralMie over current alternatives. However, I am a bit skeptical of the TAMie tool as a novelty; nevertheless, presented as part of the NeuralMie development process, it is reasonable to include it either in the main manuscript or in the typical supporting paper.

**Review data & methodology** The data set used (created) and the methodology are well described in sufficient detail, and the process followed by the authors is easy to follow. The evaluation on the performance of TAMie still depends in part on how (and with what) the comparison is made, but I consider these evaluations reasonable, and in any case not critical to this research. I do not evaluate the code provided here.

**Figures/Tables** The selection and quality of the figures and tables is also good, although probably the results of Table 2 need more information (name and optimization flags of the compiler used).

**Discussion** From my point of view, the discussion is reasonable and well organized. I appreciate that the authors describe the limitations of their new tool and the challenges of its application (these are the first steps for future improvements). However, the terminology is sometimes a bit confusing. For example, NeuralMie claims to be as accurate as a pure Mie calculation, but in other sentences it is described as "a significant improvement in accuracy". Although it is possible to understand what the authors mean from the context, perhaps they could consider alternative terminology for the first (or second) context just to add more clarity for readers not expert in aerosol modeling. Alternatively, a few words to specify the two contexts of "accuracy" may be helpful.

**Code/Datasets** The quality of presentation is good if we omit an evaluation of the source code in the review. As I commented, the source code (and datasets) doesn't have any typical README file, a simple make-file or script to help reviewers analyze/use the tool, nor any file describing the future licensing of the software.

**Specific comments**

- **Abstract** The authors comment: "This work introduces two new contributions to enable a more accurate representation of aerosol optics in atmospheric models. Enable a more accurate representation of aerosol optics in atmospheric models." However, it appears that TAMie has only been introduced to provide a faster Python-based standard Mie code, so it does not allow for a more accurate representation. In fact, TAMie is later described as being comparable in accuracy to the established Mie codes. Therefore, it seems a bit inconsistent.

- **Abstract** The authors mention: "incurring a negligible error". Usually the adjective of an error (negligible, large, small...) depends on the application. If for example the code is used to calculate by "brute force" to guess the refractive index from a given set of bulk optical parameters, the error may be more relevant. I agree that 0.08% is a fairly small error, but I usually give the user the responsibility to judge any error rating according to their application requirements. It is up to the authors to keep "negligible" or not in the summary, but my view is that "negligible" is for the user of the tool to decide.

- **Introduction** The introduction is reasonable for the scope of their manuscript, still here I include few comments. It is the final decision of the authors to take them or not in account (no requirement on my side).

- **[Line 22-23]** *"Recently, machine learning (ML), and neural networks in particular, have emerged as powerful tools for developing new, more accurate, faster, and more capable, physics parameterizations for atmosphere modeling."* This is an strong statement and probably a large number of modelers would say that ML methods could provide a more accurate or more capable parametrization but not always. Also the statement says everything at the same time: accurate, faster and capable. Is this really so general?

- **[Line 24-26]** *"Conventional parameterizations typically take the form of simplified hand-derived physical models, basic statistical models like lookup tables and linear regressions, and sometimes simply rely on expert heuristics to make decisions about the behaviour of a system."* Again I don't think that this is general, and for sure there is not a full agreement in the research community about this statement. A key set of physical parametrization are derived from fundamental physics and chemistry theories. Just two examples (but there are more): many relations of (cloud) microphysics are derived from Molecular Physics and Thermodynamics, also radiative transfer parametrization are not basic statistical models, lookup tables, linear regressions, expert heuristics or simplified hand-derived physical models (despite specific assumptions/datasets used in current radiative transfer schemes could be improved using ML methods)

- **[Line 29-31]** Do the authors think that everything is positive about the use of ML emulators here, or there are some trade-off to consider that could be worth to mention?

- **[Line 41-43]** Just note that several climate models have aerosols that are not coated aerosols but an internal mixture of an aerosols with inclusions of other species. They usually redefine the refractive index using Maxwell Garnett mixing rule [2]

- **[Line 70-75]** Here, I would be cautious, although the authors' evaluations are reasonable. First, the comparison has been done with two specific Fortran Mie codes. Numba probably is correctly setting the best performance for the LVVM tool chain for the TAMie.py code, but this comparison would need to be sure that the optimization flags of the compiler are optimal for the comparison. The performance improvement of TAMie.py with respect to PyMieScatt code is remarkable even without the use of Numba. The general value of PyMieScatt more than speed is the flexibility of their API and very good documentation, two aspects that are not present in TAMie code.

- **Sections 2. to 3.2** In this sections the authors implement Mie scattering methods in a Python tool called TAMie.py, since the definitions and equations are standard and well known, and the

presentation is straightforward I see no need for any changes in these sections since the authors have been clear on the origin and references for these definitions/equations.

- **Table 2**, as I already commented, would be good to be specific of the Fortran compiler and the optimization flags used to have a more robust performance comparison. On the other side, in drawing conclusions from Table 2, I think it should be remembered that the performance is comparable "to these Fortran codes" and is an improvement over Python's PyMieScatt code. Probably, the conclusions are fairly general, but with the evidence provided the readers cannot be sure if there is other Fortran/C code better optimized than those used in the comparison (in particular the Fortran codes used seems to be for the references at least two or three decades old). For the authors information, and thinking in future developments there a project that collects Mie codes: List of Mie codes.

- **Equation (20) and the discussion in lines [238-240]** There are modal aerosols where include internally and externally mixture of aerosols. If this something considered when the NeuralMie was developed?

- **Figure 2**. These are interesting results to undertand how increase the complexity of the model it not always improving the accuracy. Still in the case of the core-shell it seems to have an asymtopic decay. In this figure if the two branches have an structural difference they can be represented with different colors. If I understand well this figure is specific of $k_e$ because the Table 6 gives for it the higher errors. However, it the authors have results analogous to Figure 2 for the other parameters $\omega$ and $\bar{g}$ are they similar?

**Final Comment**

As the authors correctly explain, "the main limitations are not necessarily specific to the machine learning model and originate from simplifying assumptions about particle shapes and particle size distributions". This means that the practical improvements provided by the NeuralMie emulator still have the limitations due to the assumptions of the Mie solution (spherical and homogeneously coated particles). One implication is that comparison with observations will remain a challenge, as the emulator provides higher resolution in the parameters with the Mie approximation is based on, but is still using an approximation that has varying degrees of accuracy to represent each type of actual observed aerosols. Even in this case, to the extent that the refractive index estimated from observations is based on inverse methods using the Mie scattering approximation, it seems to me that the emulator may have practical utility.

But possibly the most immediate utility is its application to model comparison. If we look at the results on aerosol diversity *mass extinction efficiency* between climate models, specifically with respect to mineral dust, see reference [1], the use of these emulators may provide additional information that explains or helps to better trend these differences, as still most of them are based on Mie calculations.

As far as the code is concerned, I have not carried out an exhaustive review. For this, a README would be necessary/useful, possibly accompanied by make files. I also recommend to always include a file called LICENSE with the chosen licence and the name of the authors/institution. In a practical way, more detailed headers can also increase the usability and recognition of the code in other projects, even for relatively simple code such as TAMie.py core-suite (about 200 lines).

**References**

[1] R. Checa-Garcia et al. "Evaluation of natural aerosols in CRESCENDO Earth system models (ESMs): mineral dust". In: *Atmospheric Chemistry and Physics* 21.13 (2021), pp. 10295–10335. DOI: 10.5194/acp-21-10295-2021. URL: https://acp.copernicus.org/articles/21/10295/2021/.

[2] M. Sand et al. "Aerosol absorption in global models from AeroCom phase III". In: *Atmospheric Chemistry and Physics* 21.20 (2021), pp. 15929–15947. DOI: 10.5194/acp-21-15929-2021. URL: https://acp.copernicus.org/articles/21/15929/2021/.

---

## Author Comment (AC1)

We sincerely thank both reviewers for their detailed comments and feedback. We have implemented several changes to the manuscript in response to this feedback and believe it is improved over the original submission. The primary change is the expansion of the discussion section with some consideration of computational costs. We have also fixed several small issues throughout the paper and added detail and clarification in places in response to Reviewer 2. Finally, we have added more comments and a detailed readme file to the NeuralMie and TAMie code explaining how to use it. The code is now published at: https://github.com/pnnl/NEURALMIE and has been archived on Zenodo.

**Reviewer #1 (Peter Ukkonen):**

We agree that the computational cost of the new parameterization is a major concern that should be addressed in the paper (and will impact the parameterization's adoption). We have benchmarked the Fortran versions of the neural networks and found that calls to the homogeneous sphere model take about 3μs on a single core and calls to the core-shell model take 6.3μs (on the same hardware the Mie code was tested on). We have also now run two short test simulations with E3SM and a control simulation using the neural network to perform the aerosol optics calculations and have added new discussion to the paper (lines 510-529) regarding computational requirements. In short, we found that the neural networks can be used in E3SM simulations (the simulations run stably and have plausible aerosol radiative effects), but the homogeneous sphere model increased runtime by about 2x while the core-shell model increased runtime by about 4x. This is fast enough to perform single, aerosol-focused experiments perhaps, but too slow to serve as the default optics parameterization. We are exploring several options for accelerating the neural network solution (with 1 and 2 below suggested by the reviewer) and will document the results in a separate manuscript documenting the implementation and evaluation of the emulator in E3SM:

1) Batched inference. The neural network needs to be called once for each wavelength and aerosol mode combination, which amounts to 120 times per grid-cell and level for MAM4, so there is potential to increase efficiency here.

2) Predicting all wavelengths in a single inference step. The neural network can be re-configured to output values for each wavelength band used in E3SM. If the size of the ANN does not increase substantially that would result in a large speedup. We would like to avoid this type of solution because it is less generalizable and would be specific to the wavelengths used in E3SM (though these are shared by several other ESMs at least). Another difficulty with this solution is that the refractive index is wavelength dependent.

3) Reducing model size: we chose model configurations based on Figure 2 in the paper, and reducing model size increases error, but the relationship is non-linear. Reducing the model size by a factor of 5 increases error by a factor of 2 and reducing model size by a factor of 10 increases error by a factor of 4. Because the MAE of the original neural networks is so low it may be acceptable to use a smaller model. We may also be able to reduce model size through pruning or using a different model architecture.

4) Using multiple smaller neural networks. Currently we use the same emulator for all wavelengths. In the longwave, only absorption needs to be calculated, however. Also, some of our experiments have indicated that the specific extinction can be calculated accurately with a smaller network if it is handled by a separate network from the single scattering albedo and asymmetry parameter.

5) Only calling the neural network in cases where there is a significant amount of aerosol: currently the model is run for every grid-cell, level, time-step, wavelength-band, and aerosol mode (when Rayleigh or geometric approximations cannot be used). This means it is called at locations and times with very low aerosol loading and, in these cases, we could fall back to the existing parameterization.

Also, Reviewer 1's comment noted that we indicate there are 32 wavelength bands used in EAM in the paper. This was a mistake and there are actually 30. The manuscript has been updated to reflect this.

**Reviewer #2:**

We have made several changes to the manuscript in response to Reviewer 2's feedback:

Reviewer 2 noted that a README and LICENSE file were not provided with the code. The code is now publicly available from https://github.com/pnnl/NEURALMIE under a BSD-2 license. We have added a readme that describes how to use both TAMie and NeuralMie. There are also scripts that provide examples of how to load and inference the trained neural networks (with appropriate input and output scaling) and how to use the Rayleigh approximation code for small particles. We have also released the Zenodo repositories originally provided privately to the reviewers to the public. This information has been added to the code/data availability section of the paper.

We have clarified statements about the accuracy of NeuralMie, to make it clear that it is more accurate than the parameterization currently used in E3SM but not more accurate than Mie code. Also, we have clarified the use of the term "negligible" in the abstract, and have either removed or clarified such modifiers elsewhere in the paper.

We have also made some slight changes to wording and ordering in the abstract and introduction to make it clear that the main novel scientific contribution is NeuralMie while TAMie was a part of its development and is based on existing algorithms. We have included the details about TAMie's development and testing in the paper and are releasing the code because we think this software fills a gap in the currently available Mie scattering libraries and others that wish to do fast Mie scattering calculations in Python will find the code useful.

Below are responses to some of Reviewer 2's specific comments:

*[Line 22-23] "Recently, machine learning (ML), and neural networks in particular, have emerged as powerful tools for developing new, more accurate, faster, and more capable, physics parameterizations for atmosphere modeling." This is a strong statement and probably a large number of modelers would say that ML methods could provide a more accurate or more capable parametrization but not always. Also the statement says everything at the same time: accurate, faster and capable. Is this really so general?*

We have edited this statement in the paper to be a bit more conservative. We certainly do not mean to imply that all parameterizations can, or should, be replaced by ML, just that ML has potential to replace some with solutions that are more accurate, faster, **and/or** (changed from "and") more capable. In this case, NeuralMie is more accurate and more capable, but slower than the existing parameterization for example.

*[Line 24-26] "Conventional parameterizations typically take the form of simplified hand-derived physical models, basic statistical models like lookup tables and linear regressions, and sometimes simply rely on expert heuristics to make decisions about the behaviour of a system." Again I don't think that this is general, and for sure there is not a full agreement in the research community about this statement. A key set of physical parametrization are derived from fundamental physics and chemistry theories. Just two examples (but there are more): many relations of (cloud) microphysics are derived from Molecular Physics and Thermodynamics, also radiative transfer parametrization are not basic statistical models, lookup tables, linear regressions, expert heuristics or simplified hand-derived physical models (despite specific assumptions/datasets used in current radiative transfer schemes could be improved using ML methods)*

We have expanded this statement to be broader and open-ended, and note that ML should be seen as a potentially powerful addition to the existing set of methods.

*[Line 29-31] Do the authors think that everything is positive about the use of ML emulators here, or there are some trade-off to consider that could be worth to mention?*

There are certainly tradeoffs, and we added a sentence to this paragraph to make this clear. We do not want to provide a full audit of the pros and cons of using neural networks for parameterizations and physics emulation here though, so as not to overburden the introduction. We have also added a more detailed discussion of the computational expense of NeuralMie to the Discussion section (lines 510-529) which is a tradeoff for this particular ML solution.

*[Line 41-43] Just note that several climate models have aerosols that are not coated aerosols but an internal mixture of an aerosols with inclusions of other species. They usually redefine the refractive index using Maxwell Garnett mixing rule [2]*

We added a note here (line 43)

*[Line 70-75] Here, I would be cautious, although the authors' evaluations are reasonable. First, the comparison has been done with two specific Fortran Mie codes. Numba probably is correctly setting the best performance for the LVVM tool chain for the TAMie.py code, but this comparison would need to be sure that the optimization flags of the compiler are optimal for the comparison. The performance improvement of TAMie.py with respect to PyMieScatt code is remarkable even without the use of Numba. The general value of PyMieScatt more than speed is the flexibility of their API and very good documentation, two aspects that are not present in TAMie code.*

Information about the Fortran compiler used for the comparison of TAMie to Fortran codes was added on line 180.

We have also added comments to Section 2.2 noting some of the drawbacks of TAMie and limitations of the comparison, specifically:

- Other, more modern, compiled Mie scattering codes exist.

- Faster run times could likely be achieved by the Fortran codes using appropriate compiler optimization flags.

- While TAMie is substantially faster than PyMieScatt, PyMieScatt has many more features.

*Equation (20) and the discussion in lines [238-240] There are modal aerosols where include internally and externally mixture of aerosols. If this something considered when the NeuralMie was developed?*

NeuralMie is flexible and can handle internal or external mixing assumptions (or a combination) by doing multiple inferences. For instance, to run NeuralMie with MAM4, which assumes internal mixing within four separate modes, the model will be inferenced once per mode. For fully external mixing it would be inferenced once per aerosol species. We added a note on MAM4's mixing assumptions on line 42, and have added a comment about how NeuralMie handles this on line 323.

*Figure 2. These are interesting results to understand how increase the complexity of the model it not always improving the accuracy. Still in the case of the core-shell it seems to have an asymtopic decay. In this figure if the two branches have an structural difference they can be represented with different colors. If I understand well this figure is specific of ke because the Table 6 gives for it the higher errors. However, it the authors have results analogous to Figure 2 for the other parameters ω and g̅ are they similar?*

The main purpose of this figure is to illustrate the asymptotic behavior and how we chose a model architecture based on these results. We would prefer not to add complexity by coloring different layer counts (we tested 2, 3, and 4 layer networks in this work), but included this comment in the caption to explain why this separate group of models appears in the plot since we thought readers might find this curious. In any case, rather than adding more detail to this plot we added a reference to our previous work on this topic that involved a much more detailed analysis of model architectures and contains a similar plot with models separated out by number of layers, and that is based on error across all three output variables.